# Constraint of Different Knee Implant Designs Under Anterior–Posterior Shear Forces and Internal–External Rotation Moments in Human Cadaveric Knees

**DOI:** 10.3390/bioengineering12010087

**Published:** 2025-01-19

**Authors:** Saskia A. Brendle, Sven Krueger, Joachim Grifka, Peter E. Müller, William M. Mihalko, Berna Richter, Thomas M. Grupp

**Affiliations:** 1Research & Development, Aesculap AG, 78532 Tuttlingen, Germany; 2Department of Orthopaedic and Trauma Surgery, Musculoskeletal University Center Munich (MUM), Campus Grosshadern, LMU Munich, 81377 Munich, Germany; 3Department of Orthopaedics, Asklepios Klinikum, 93077 Bad Abbach, Germany; 4Campbell Clinic Department of Orthopaedic Surgery & Biomedical Engineering, University of Tennessee Health Science Center, Memphis, TN 38104, USA

**Keywords:** knee, biomechanics, cadaveric study, anterior–posterior stability, TKA design

## Abstract

Instability remains one of the most common indications for revision after total knee arthroplasty. To gain a better understanding of how an implant will perform in vivo and support surgeons in selecting the most appropriate implant design for an individual patient, it is crucial to evaluate the implant constraint within clinically relevant ligament and boundary conditions. Therefore, this study investigated the constraint of three different implant designs (symmetrical implants with and without a post-cam mechanism and an asymmetrical medial-stabilized implant) under anterior–posterior shear forces and internal–external rotation moments at different flexion angles in human cadaveric knees using a six-degrees-of-freedom joint motion simulator. Both symmetrical designs showed no significant differences between the anterior–posterior range of motion of the medial and lateral condyles. In contrast, the medial-stabilized implant exhibited less anterior–posterior translation medially than laterally, without constraining the medial condyle to a fixed position. Furthermore, the post-cam implant design showed a significantly more posterior position of the femoral condyles in flexion compared to the other designs. The results show that despite the differences in ligament situations and individual implant positioning, specific characteristics of each implant design can be identified, reflecting the different geometries of the implant components.

## 1. Introduction

In total knee arthroplasty (TKA), various implant designs are available which generate different kinematic profiles and envelopes of stability [1]. Symmetrical implants with different conformities, with or without a post-cam mechanism, are most commonly used and have been established for decades, with promising long-term results [2,3]. In comparison, the medial-pivot implant design is a more recent innovation and has not been studied in depth. With this design, the motion is guided by a highly congruent medial compartment, which behaves similarly to a ball in socket and creates a fixed medial pivot point [4]. In addition, there is a new generation of medial-stabilized implants with less medial conformity that have been designed to closely mimic the kinematic pattern of the healthy knee, with no fixed pivot point [3,5,6,7].

To date, many in vivo and in vitro studies have focused on comparing the resulting kinematics of different implant designs [8,9,10,11,12,13,14,15,16]. However, since instability remains one of the most common indications for revision after total knee arthroplasty [2,17,18,19,20], it is important to select the implant design for each patient individually, not only to meet the individual patient’s needs in terms of kinematics [9], but also to ensure an adequate envelope of stability. For this reason, it is important to understand how much stability a particular implant design provides at different flexion angles to select the appropriate design for each individual patient in a more personalized approach. To accurately compare the anterior–posterior constraint of different implant designs, it is essential to have a highly controlled environment and precise application of forces. Additionally, it is crucial to test not only the implant itself but also its behavior under clinically relevant boundary conditions that the surrounding ligament conditions will provide. While some in vivo studies allow for controlled application of forces by arthrometers [21,22,23,24,25,26], others are conducted by applying manual force [27], which has limited reproducibility. Moreover, in vivo studies do not provide the opportunity to test different implant designs within the same ligamentous situation [21,22,23,24,25,26,28]. In contrast, biomechanical in vitro studies offer a high degree of accuracy and control, as well as the ability to measure the constraint of different implant designs within the same ligamentous situation [9,29,30]. However, no studies to date have thoroughly investigated the anterior–posterior position and range of motion of the femoral condyles under anterior–posterior shear forces and internal–external rotational moments with various implant designs in the same knees [30,31,32]. Furthermore, each TKA design exhibits unique characteristics, and not all cruciate-retaining (CR), cruciate-sacrificing (CS), medial-stabilized (MS), or posterior-stabilized (PS) designs share the same properties. For this reason, it is imperative to conduct detailed analyses of each newly developed design to assist surgeons in selecting the most appropriate implant design to address the specific needs of individual patients.

Therefore, the objective of this study was to investigate the constraint of symmetrical implants with and without a post-cam mechanism, as well as an asymmetrical medial-stabilized implant design during anterior–posterior shear forces and internal–external rotational moments at various flexion angles in human cadaveric knees. The hypothesis was that the symmetrical TKA designs have the same anterior–posterior range of motion medially and laterally despite the differences in the individual ligamentous situations. We anticipated that, with the medial-stabilized TKA design, the medial condyle would exhibit less anterior–posterior translation compared to the lateral condyle, without constraining the medial condyle to a fixed position. Furthermore, we hypothesized that, with the post-cam TKA design, the femoral condyles would be located more posterior in flexion compared to the other TKA designs.

## 2. Materials and Methods

### 2.1. Specimen Preparation

Thirteen fresh-frozen human cadaveric lower right extremities (three females and ten males) were used in this study. The donors had a mean age of 67 ± 10 years and a mean body mass index of 23.3 ± 8.0 kg/m^2^. The specimens were screened for pre-existing knee disorders, surgical interventions, and other relevant pathologies. Ethical approval was obtained from the ethics committee of the Ludwig Maximilian University of Munich (No. 20-0856).

Before the experiment, each specimen was thawed for 24 h at 7 °C and then prepared for testing based on a methodology recently published by Brendle et al. [29]. The proximal and distal segments of the joint were skeletonized, while care was taken to preserve soft tissue surrounding the knee joint capsule. GOM measuring points (1.5 mm, Carl Zeiss GOM Metrology GmbH, Braunschweig, Germany) were attached to the skeletonized parts of the bones, and 3D fittings of femur and tibia were performed by aligning segmented CT scans (Mimics 24.0, Materialise, Leuven, Belgium) to 3D point clouds of each bone (ARAMIS 12M, Carl Zeiss GOM Metrology GmbH, Braunschweig, Germany). Afterwards, the 3D fitting information, containing the 3D point cloud to CT scan information, was saved and the bones were cut and embedded in custom-made aluminum pots for mounting in a six-degrees-of-freedom joint motion simulator. For this purpose, the femur was aligned with the axes of the upper actuator of the joint motion simulator and the tibia was embedded at 0° flexion.

After initial preparation, the specimens underwent cruciate-sacrificing total knee arthroplasty [9] by an experienced knee surgeon using the oneKNEE^®^ TKA system (Aesculap AG, Tuttlingen, Germany), which allows the use of a femoral component with or without a post-cam mechanism (PS or CR/CS) with the same bone cuts. Furthermore, the system has the option to utilize a symmetrical tibial inlay with medium conformity without a post-cam mechanism (CR/CS), a symmetrical tibial inlay with medium conformity and a post-cam mechanism (PS), or an asymmetrical medial-stabilized tibial inlay with higher conformity medial and lower conformity lateral (MS) fixed to the same tibial component (Figure 1). The components were implanted using mechanical alignment and 0° tibial slope. After implantation, the position of the implants in relation to the bones was measured (ARAMIS 12M, Carl Zeiss GOM Metrology GmbH, Braunschweig, Germany) [9].

### 2.2. Experimental Testing

Testing was performed on a six-degrees-of-freedom joint motion simulator (VIVO, Advanced Mechanical Technologies Inc., Watertown, MA, USA), which allows independent control of each degree-of-freedom in either force or displacement mode. The forces and motions are expressed in accordance with the Grood and Suntay conventions [33]. After mounting the specimen, the absolute joint position of the specimen was transferred to the joint motion simulator by projecting the previously generated 3D fitting information of the segmented CT scans onto the residual bones using the remaining measuring points. In this way, the 3D information of the complete femur and tibia were available even after the bones were cut. Subsequently, each specimen was subjected to dynamic testing. In order to characterize the constraint of the implant designs in each knee, cyclic anterior–posterior shear forces of ±80 N and internal–external rotation moments of ±5 Nm were sequentially applied as a ramp profile for four cycles at 0, 30, 45, 60, and 90° of flexion while maintaining an axial compression force of 200 N and all other forces and moments at 0 N/Nm. The applied loads were chosen to reflect those used in clinical assessment of joint laxity [34] and were within the range of forces used in comparable studies [35,36,37]. Each loading protocol was applied to the CR/CS, MS, and PS TKA designs. Thereby, the CR/CS and MS implant designs were tested in a randomized order in each specimen. The PS implant design was always tested last. For all conditions, the knee capsule was opened using a medial parapatellar approach and closed with surgical sutures (Number 1 Vicryl, B. Braun, Melsungen, Germany). During testing, the relative position of femur and tibia were tracked by the joint motion simulator. The passive tension of the patella tendon was simulated by a spring sutured to the quadriceps tendon with an increasing force of up to 50 N at 90° flexion. Furthermore, the specimens were kept moist with sodium chloride solution to mitigate the effects of tissue drying during testing. Figure 2 illustrates the entire testing process.

### 2.3. Data Analysis

The data from a single cycle of each experiment were evaluated with custom MATLAB scripts (Version R2023a, MathWorks Inc., Natick, MA, USA) [29]. The flexion axis and the centers of the medial and lateral condyles of the femoral component were projected onto the tibial plane at various flexion angles (0, 30, 45, 60, and 90°) at the first time reaching the maximum or minimum force or moment, representing the positions of the femoral condyles on the tibial plateau (Figure 3). The positions were normalized to the anterior–posterior width of the implanted tibial plateau of the respective tibia to minimize the effect of the implant component size. Thus, irrespective of the different tibia size, 0 and 1 correspond to the most posterior and most anterior position on the tibial plateau, respectively, as illustrated in Figure 3. Furthermore, the normalized anterior–posterior range of motion of the medial and lateral condyles was calculated for all implant designs as the difference between the positions at maximum and minimum force or moment.

Statistical analyses were performed in Minitab (Version 21.2, Minitab GmbH, Munich, Germany). Wilcoxon-signed rank tests [38] were used to compare the position at maximum anterior/posterior shear force and internal/external rotation moment, respectively, between the different TKA designs pairwise at various flexion angles. Furthermore, the normalized anterior–posterior range of motion was compared between the medial and lateral condyles, as well as between the different TKA designs at various flexion angles during anterior–posterior (AP) shear forces and internal–external (IE) rotation moments. The level of significance was set at *p* ≤ 0.05. Some specimens exceeded the range of motion of the joint motion simulator during testing and were therefore excluded from data analysis (*n* = 2 for AP shear forces and *n* = 3 for IE rotation moments).

## 3. Results

### 3.1. Anterior–Posterior Shear Forces

Figure 4 shows the projection of the flexion axis and the centers of the medial and lateral condyles of the femoral component onto the tibial plane during anterior–posterior shear forces applied on the tibia at 45° flexion with a CR/CS, MS, and PS TKA design exemplary for one specimen. The posterior translation of the femoral condyles at maximum anterior directed force (positive) is colored in blue, whereas the anterior translation of the femoral condyles at maximum posterior directed force (negative) is colored in orange. With the CR/CS and the PS TKA designs, the femoral condyles exhibited a parallel translation with less anterior displacement for the PS TKA design at maximum negative force. In contrast, with the MS TKA design, the medial condyle showed less posterior translation than the lateral condyle, due to the higher constraint of the medial compartment.

Figure 5 shows the normalized medial and lateral anterior–posterior range of motion during anterior–posterior shear forces with the CR/CS, MS, and PS TKA designs for all specimens. Furthermore, it illustrates the anterior–posterior positions of the medial and lateral femoral condyles at maximum anterior and posterior shear forces, respectively, with the different TKA designs at various flexion angles for all specimens with boxplots on a normalized tibia. Boxplots include the median AP positions, the first and third quartiles, and the range. Outliers are displayed as dots, and significant differences between the values (*p* ≤ 0.05) are marked with an asterisk. In addition, median values and statistical significances are presented in Table A1 for the AP range of motion, and in Table A2 for the AP positions at maximum anterior and posterior shear forces, respectively.

As already observed for the exemplary specimen, the CR/CS design demonstrated a greater medial AP range of motion compared to the MS and PS designs, with significant differences at all flexion angles. Furthermore, the MS and PS designs significantly differed at 0, 30, and 90° flexion, with a smaller medial AP range of motion for the PS design at 90°. Laterally, the PS design had the smallest AP range of motion compared to the other designs and significantly differed from the CR/CS design at all flexion angles. For all designs, the smallest medial and lateral AP ranges were observed at 0° flexion, with the MS design showing a significantly smaller medial AP range of motion than the symmetrical designs. For the MS design, the medial AP range of motion was generally smaller than the lateral AP range of motion with significant differences at 45, 60, and 90° flexion. The CR/CS and PS designs also exhibited a smaller medial AP range of motion at 30 and 45° flexion but showed almost identical medial and lateral ranges at 60 and 90° flexion. Furthermore, the variance between the individual specimens was greatest in mid-flexion.

During anterior directed force on the tibia, the medial condyle was positioned significantly more anterior with the MS design than with the CR/CS and PS designs at all flexion angles. From 30° of flexion, the medial and lateral condyles were positioned at approximately the same height with the symmetrical designs. In contrast, with the MS design, the medial condyle was positioned more anteriorly than the lateral condyle. For all TKA designs, the position of the condyles shifted posteriorly at higher flexion angles. At 90° flexion, the femoral condyles were positioned significantly more posterior with the PS design than with the CR/CS and MS designs.

During posterior directed force on the tibia, the lateral condyle was positioned more anterior than the medial condyle with all TKA designs at 0 and 30° of flexion, and were approximately at the same height at 45° flexion. From 30°, both femoral condyles were positioned significantly more posterior with the PS design than with the CR/CS design. In addition, from 45° of flexion, the femoral condyles were positioned significantly more posterior with the PS design compared to the MS design. Furthermore, the position of the CR/CS and MS designs revealed significant differences at 30, 45, and 60° flexion, with a more posterior position of the MS design. With the PS design, the femoral condyles shifted posteriorly with flexion. In contrast, the median position of the femoral condyles remained approximately the same throughout the range of flexion with the CR/CS and MS designs.

### 3.2. Internal–External Rotation Moments

Figure 6 shows the projection of the flexion axis and the centers of the medial and lateral condyles of the femoral component onto the tibial plane during internal–external rotation moments applied on the tibia at 45° flexion with a CR/CS, MS, and PS TKA design exemplary for one specimen. The internal rotation of the femoral condyles at maximum external rotation moment (positive) is shown in blue, whereas the external rotation of the femoral condyles at maximum internal rotation moment (negative) is shown in orange. With the CR/CS and the PS TKA designs, the medial and lateral condyles exhibited approximately the same anterior–posterior range of motion during internal–external rotation moments, resulting in a rotation around the center of the tibia with a small medial offset. In contrast, with the MS TKA design, the medial condyle showed less anterior–posterior translation compared to the lateral condyle, resulting in a rotation around the medial compartment of the tibia.

Figure 7 shows the normalized medial and lateral anterior–posterior range of motion during internal–external rotation moments with the different TKA designs for all specimens. In addition, it presents the anterior–posterior positions of the medial and lateral femoral condyles at maximum internal and external rotation moments, respectively, with a CR/CS, MS, and PS TKA design at various flexion angles for all specimens with boxplots on a normalized tibia. Boxplots include the median AP positions, the first and third quartiles, and the range. Outliers are displayed as dots, and significant differences between the values (*p* ≤ 0.05) are marked with an asterisk. Additionally, median values and statistical significances are presented in Table A3 for the AP range of motion and in Table A4 for the AP positions at maximum internal and external rotation moments.

As observed for the exemplary specimen, the symmetrical designs exhibited approximately the same AP range of motion medially and laterally, whereas the MS design showed smaller ranges medially with significant differences at 60 and 90° flexion. Furthermore, the MS design showed a significantly smaller medial AP range of motion compared to the CR/CS design at all flexion angles. In contrast, the lateral condyle showed a significantly larger AP range of motion with the MS design compared to the CR/CS and PS designs at 30, 45, 60, and 90° flexion. For all designs, the smallest medial and lateral AP ranges were observed at 0° flexion.

During external rotation moment, the medial condyle was positioned significantly more anterior with the MS design than with the CR/CS and PS designs at all flexion angles. Furthermore, with the PS design, the position of the medial condyle differed significantly from that obtained with the CR/CS design at 0, 45, 60, and 90° of flexion, with a more anterior position at 0° flexion and a more posterior position at 45, 60, and 90° of flexion. The lateral condyle was located more posterior with the PS design and differed significantly from the MS (all flexion angles) and the CR/CS (45, 60, and 90° flexion) designs. In addition, the median position of the medial condyle shifted posteriorly through the range of flexion with all TKA designs, whereas the median position of the lateral condyle stayed nearly unchanged with the CR/CS and MS designs and was considerably more posterior with the PS design.

During internal rotation moment, both femoral condyles were positioned significantly more posterior with the MS design compared to the CR/CS design at 45, 60, and 90° flexion. Furthermore, the median position of the lateral condyle shifted posteriorly through the range of flexion for all TKA designs. In contrast, the median position of the medial condyle remained almost the same with the CR/CS and MS designs but was considerably more posterior with the PS design. At 60° flexion, both femoral condyles were significantly more posterior with the PS design compared to the CR/CS design. In addition, at 90° flexion, the femur was positioned significantly more posterior with the PS design compared to both the CR/CS and MS designs.

## 4. Discussion

To gain a better understanding of how an implant will perform in vivo and select the most appropriate implant design to address a patient’s individual needs, it is crucial to evaluate the differences in constraint not only for isolated implant components, but within clinically relevant ligament and boundary conditions [39]. Therefore, the objective of this study was to investigate the constraints of three different implant designs out of a newly developed comprehensive knee platform during anterior–posterior shear forces and internal–external rotation moments at various flexion angles in human cadaveric knees. We confirmed the first hypothesis that the symmetrical TKA designs have the same anterior–posterior range of motion medially and laterally. During both anterior–posterior shear forces and internal–external rotation moments, the medial and lateral anterior–posterior ranges showed no significant differences for the CR/CS and PS TKA designs, despite the differences in the individual ligamentous situations. In contrast, with the medial-stabilized TKA design, the medial condyle exhibited significantly less anterior–posterior translation compared to the lateral condyle, but still allowed sliding and did not constrain the medial condyle to a fixed position when subjected to anterior–posterior shear forces, confirming the second hypothesis. In addition, as expected, the femoral condyles were located significantly more posterior with the post-cam TKA design compared to the other TKA designs in flexion.

Each implant design has a different constraint based on its geometry and associated behavior when subjected to different forces and moments. In this study, the constraint under anterior–posterior shear forces and internal–external rotational moments was investigated. This constraint comprises several aspects: the anterior–posterior range of motion and the maximum possible anterior and posterior translation of the medial and lateral femoral condyles on the tibial plateau. Clinically, it is important to understand the constraint of different implants, as it essentially defines the envelope for kinematics. However, the individual kinematics of different patients may vary within this envelope [9]. Ideally, the implant design should allow the individuals physiological kinematics while providing stability. An excessive highly constrained implant may compromise the physiological kinematics of a particular patient, whereas a constraint that is too low may cause variations in translation, resulting in an unstable situation in some patients. Thus, each patient has individual needs, and greater stability does not necessarily correlate with higher patient satisfaction [22,24,25,26]. In the following paragraphs, the various aspects of the implant constraints are discussed.

For all designs, the smallest anterior–posterior range of motion was found in extension, with a considerable increase at 30° flexion. Due to the normalized anterior–posterior range of motion and the specific implant designs used in this study, the results cannot be directly compared with those of other studies. However, previous studies also found a significant increase in AP laxity between 0° and 30° of flexion in the native condition, as well as with different CR, CS, and ultracongruent (UC) designs [27,30]. Furthermore, Shalhoub et al. [40] and Minoda et al. [41] showed that the tibiofemoral gaps significantly increase until 30° of flexion, which also represents an increase in laxity. Therefore, these findings reflect the influence of the natural ligamentous situation of the knee in mid-flexion and demonstrate the importance of evaluating the differences in constraint not only for isolated implant components but within real ligamentous supported conditions. In motion, mid-flexion instability can lead to phenomena such as paradoxical anterior sliding of the femur. However, this was not observed for the TKA designs of the present study [9].

For both symmetrical designs, there were no significant differences between the medial and lateral AP range of motion. However, during AP shear forces, both femoral condyles exhibited a considerably higher AP range of motion with the CR/CS design than with the PS design at all flexion angles, but comparable ranges until 45° flexion during internal–external rotation moments. With the MS design, the AP range of motion of the lateral condyle was similar to the CR/CS design during anterior–posterior shear forces but was higher during internal–external rotation moments. In contrast, the medial condyle exhibited a smaller AP range of motion with the MS design than with the CR/CS design during both anterior–posterior shear forces and internal rotation moments. This is attributed to the greater conformity provided by the steeper anterior and posterior ramps of the medial tibial compartment in the MS design. Nevertheless, the MS design still allowed sliding of the femoral condyle on the tibial plateau. This is a major difference to previous medial-stabilized and medial-pivot designs, which show almost no movement of the medial condyle during various activities and therefore create a fixed medial pivot point with minimal variation between patients due to the high conforming medial compartment of the tibia [28].

When analyzing the maximum posterior translation of the femoral condyles under various conditions, it could be observed that, during both anterior shear forces and external rotation moments, the MS design allowed less posterior translation of the medial condyle compared to the CR/CS and PS designs, and was therefore positioned more anteriorly at all flexion angles. This can be explained by the steeper posterior ramp of the medial compartment of the MS inlay. In contrast, as a result of the flatter posterior ramp of the lateral compartment of the MS design, the lateral condyle showed a higher posterior translation compared to the CR/CS design during internal–external rotation moments. In flexion, both femoral condyles achieved the most posterior position with the PS design, due to the intervention of the post-cam mechanism. Furthermore, it can be observed that the PS design showed the smallest variation. This is probably due to the higher guidance of the PS design compared to the CR/CS and MS designs. For one specimen, the lateral femoral condyle was generally positioned more posteriorly than observed for the other specimens, during both anterior and posterior shear forces and internal and external rotation moments with all implant designs. Therefore, outliers exist for the maximum anterior and posterior positions, but not for the anterior–posterior range of motion.

Analysis of the maximum anterior translation of the femoral condyles showed that the CR/CS design allowed the greatest anterior translation of the medial condyle during both posterior shear forces and internal rotation moments. In comparison, with the MS design, the medial condyle showed considerably less anterior translation and was therefore located more posterior than with the CR/CS design at all flexion angles. In a previous study, Kour et al. [28] investigated the positions of the medial and lateral condyles of three different TKA designs (CR, MS, and PS) on the tibial plateau during activities of daily living using mobile biplane fluoroscopy. In addition to the high constraint of the medial compartment of the tibia in the MS design, they found that the lateral condyle shifted more anterior with the MS design compared to the PS and CR designs during some of the activities. This is not expected with the MS design tested in this study. The most anterior position of the lateral condyle was similar for the CR/CS and MS designs during external rotation moments, and slightly more posterior for the MS design when subjected to posterior shear forces. Thus, the lateral compartment of the MS design has the same constraint as the CR/CS design in the anterior direction and therefore provides the same stability. A similar behavior is expected during activities of daily living, since the envelope of kinematics is prescribed by the constraint. With the PS design, the femoral condyles were positioned more posterior compared to the other designs, especially in flexion, as the post-cam mechanism forces the femur to translate posteriorly with flexion and prevents anterior translation when subjected to posterior shear forces or internal–external rotation moments. A study by Scott et al. [23] revealed a greater mid-flexion laxity characterized by a higher anterior translation of the femur in patients with PS implants than in patients with MS implants, as the post-cam mechanism used for stabilization in PS implants usually only intervenes at flexion angles beyond 45° [42] and does not prevent anterior translation of the femoral component prior to that. However, this could not be observed in the present study due to an earlier post-cam intervention as for most of the PS designs. This demonstrates that not every design of the same type has identical characteristics. Consequently, it is crucial to characterize each newly developed design, as each one has a unique geometry and therefore specific features that are essential to understand. To gain a broader understanding of the different implant designs and the influence of various factors on the overall performance and stability, future studies should also investigate the constraint when using different alignment techniques, soft-tissue balance, and tibial slope.

To the best of our knowledge, this is the first study that has compared the constraint of different implant designs in the same knees in a highly controlled environment. By examining the different TKA designs in the same knees, the variability between different cohorts can be eliminated. Furthermore, precise application of forces along reliable axes avoids false conclusions due to variations in the treatment of the different conditions and ensures a valid comparison of the different TKA designs [9,29]. Nevertheless, several limitations should be taken into account when interpreting the results of the present study. First, this study investigated only a small number of human cadaveric specimens, which may have different soft tissue characteristics from living patients. However, comparing different TKA designs in the same knees under highly controlled conditions is not possible in vivo. Second, physiological muscle loading was not applied, and the patellar mechanism was only passively simulated. Third, the constraint of the TKA designs was only investigated until 90° of flexion, and the results at higher flexion angles may differ. However, most of the activities of daily living are covered with flexion angles of up to 90° [43]. Fourth, the projection of the centers of the medial and lateral femoral condyles onto the tibial plane was used to analyze the anterior–posterior position and range of motion. This is a valid and widely used method to approximate the tibiofemoral contact pattern, especially for single-radius implant designs such as the CR/CS femoral component in the present study. However, the PS femoral component slightly changes its radius of curvature in flexion. For a better comparison of the position of the femoral condyles between the different implant designs in the same ligamentous situation, we chose to always project the CR/CS centers onto the tibial plane. This results in small inaccuracies when approximating the tibiofemoral contact pattern of the PS design in flexion. However, since the present study only investigated flexion angles up to 90°, the error should be limited and not change the general characteristic behavior [44]. Fifth, the PS TKA design was always tested last since the femoral component had to be replaced. For this reason, time-dependent effects cannot be completely excluded. However, the limited number of tests and the short test duration should mitigate these effects [45]. Finally, this study was performed using oneKNEE^®^ TKA components, and the results may not be applicable to other TKA designs.

## 5. Conclusions

This study investigated the constraints of three different implant designs from a newly developed comprehensive knee platform during anterior–posterior shear forces and internal–external rotation moments at various flexion angles in human cadaveric knees. It was found that, despite the differences in ligament situations and individual implant positioning, specific characteristics of the individual implant designs can be identified, reflecting the different geometries of the implant components. The results help to understand how much stability a particular implant design provides at different flexion angles under clinically relevant conditions and may therefore assist in selecting the most appropriate implant design to address the specific needs of individual patients.

## Figures and Tables

**Figure 1 bioengineering-12-00087-f001:**
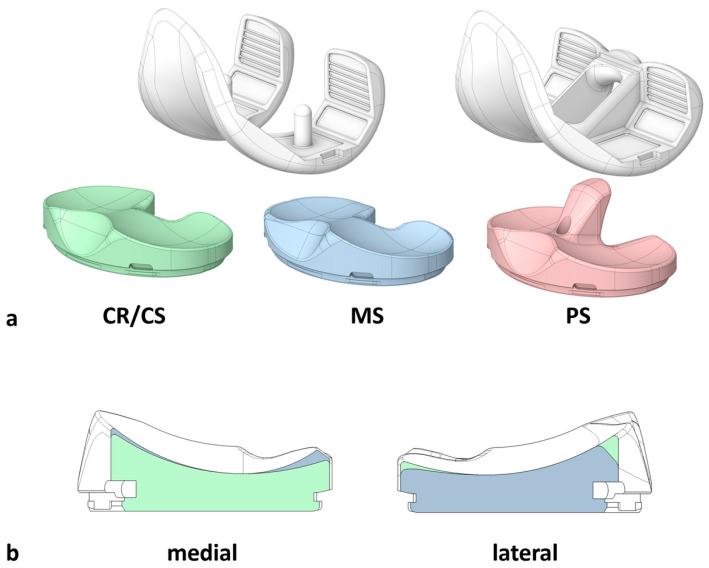
(**a**) Illustration of the total knee arthroplasty (TKA) components. The CR/CS femoral component (left) can be used with a symmetrical inlay with medium conformity without a post-cam mechanism (CR/CS, green) or an asymmetrical medial-stabilized inlay with higher conformity medial and lower conformity lateral (MS, blue). The PS femoral component (right) can be used with a symmetrical inlay with medium conformity and a post-cam mechanism (PS, red). All inlay designs can be fixed to the same tibial component, which is not illustrated in this Figure. (**b**) Schematic illustration of a cross-section through the dwell point of the medial and lateral compartments of the CR/CS (green) and MS (blue) inlay designs. The medial compartment of the MS design has steeper anterior and posterior ramps compared to the CR/CS design. The lateral compartment of the MS design has a flatter posterior ramp compared to the CR/CS design. In the shown cross-sections, the PS design has the same characteristics as the CR/CS design and is therefore not presented separately.

**Figure 2 bioengineering-12-00087-f002:**
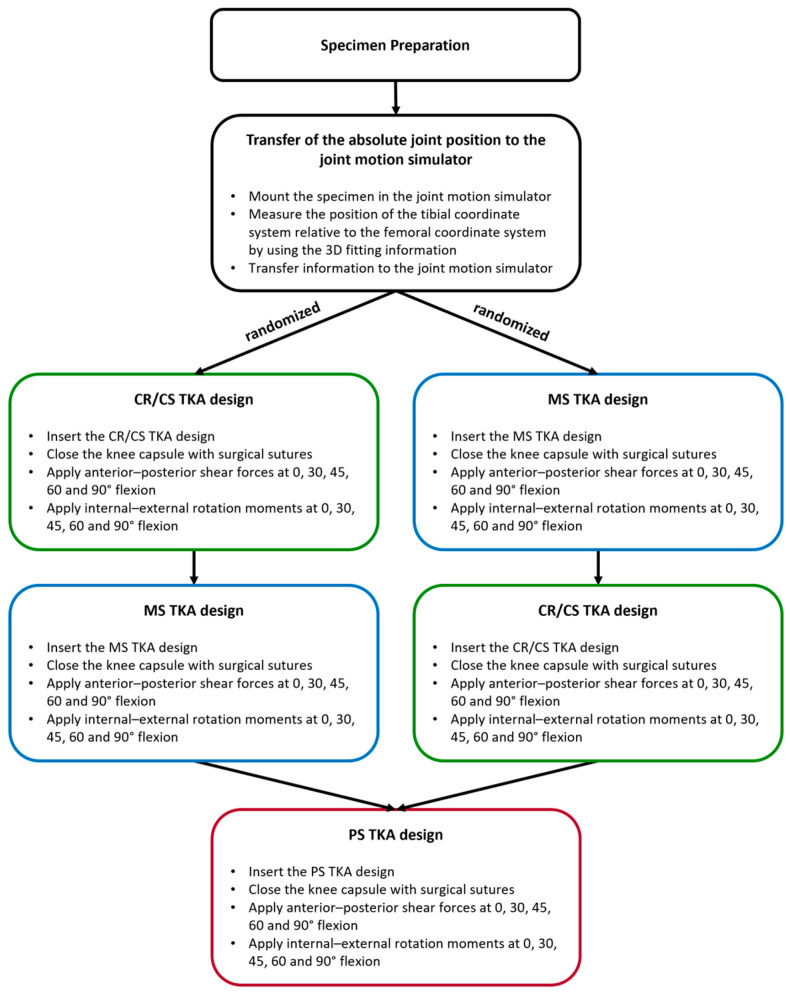
Illustration of the key steps of the testing process. TKA = total knee arthroplasty. CR/CS = symmetrical TKA design without a post-cam mechanism. MS = asymmetrical medial-stabilized TKA design. PS = symmetrical TKA design with a post-cam mechanism.

**Figure 3 bioengineering-12-00087-f003:**
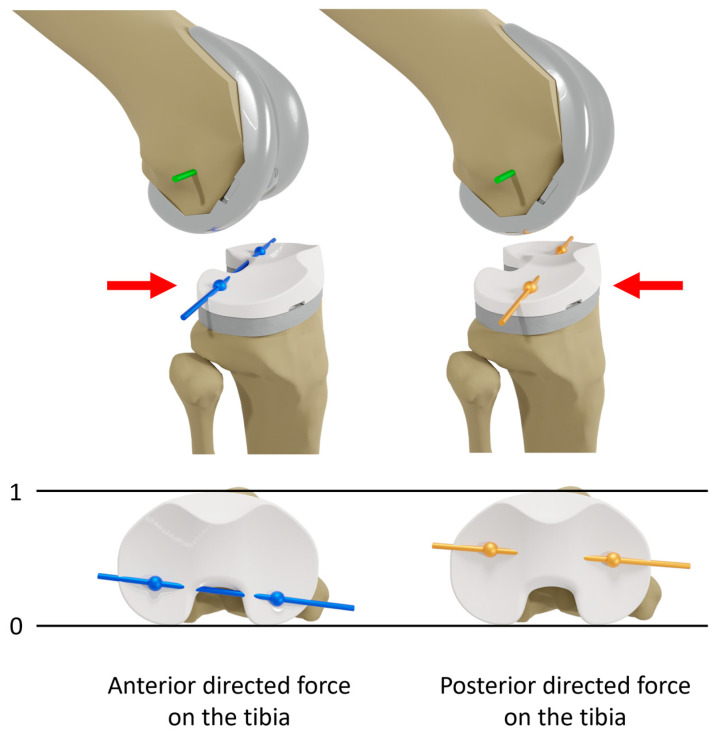
Illustration of the projections of the flexion axis (**green**) and the centers of the medial and lateral condyles of the femoral component on the tibial plane at 45° flexion under maximum anterior (**blue**) and posterior (**orange**) directed force on the tibia. Red errors indicate the direction of shear force applied on the tibia. The projections represent the positions of the medial and lateral femoral condyles on the normalized tibial plateau. Irrespective of the different tibia sizes, 0 and 1 correspond to the most posterior and most anterior position on the tibial plateau, respectively.

**Figure 4 bioengineering-12-00087-f004:**
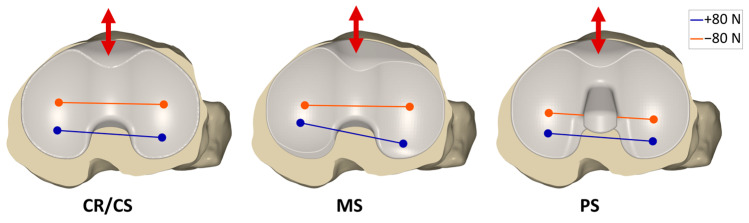
Projections of the flexion axis and the centers of the medial and lateral condyles of the femoral component onto the tibial plane with ±80 N anterior (+) and posterior (−) shear force applied on the tibia at 45° flexion, showing the condylar motion with a CR/CS, MS, and PS TKA design exemplary for one specimen. The colors represent the respective force on the tibia. Blue = 80 N; orange = −80 N.

**Figure 5 bioengineering-12-00087-f005:**
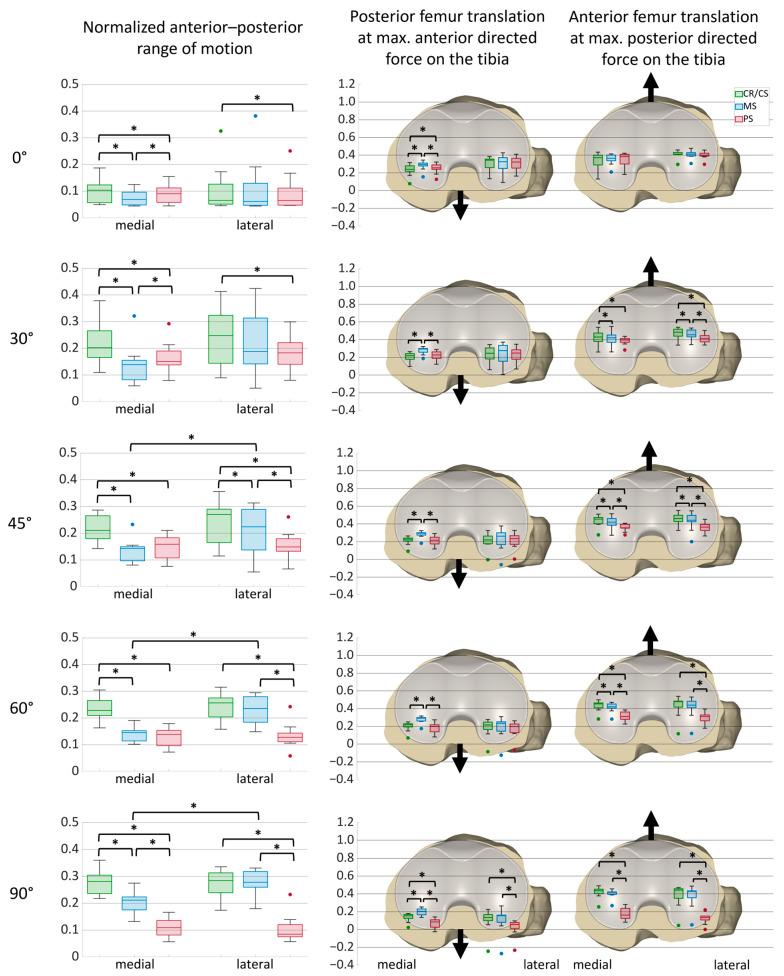
Anterior–posterior range of motion during anterior–posterior shear forces at different flexion angles and anterior–posterior positions of the medial and lateral femoral condyles at maximum anterior and posterior shear forces, respectively, at different flexion angles on a normalized tibia (*n* = 11). The anterior–posterior range of motion and the positions are normalized to an anterior–posterior tibia width of 1. Significant differences are marked with an asterisk (*p* ≤ 0.05). Posterior femur translation = anterior directed force on the tibia; anterior femur translation = posterior directed force on the tibia. Green = CR/CS; blue = MS; red = PS.

**Figure 6 bioengineering-12-00087-f006:**
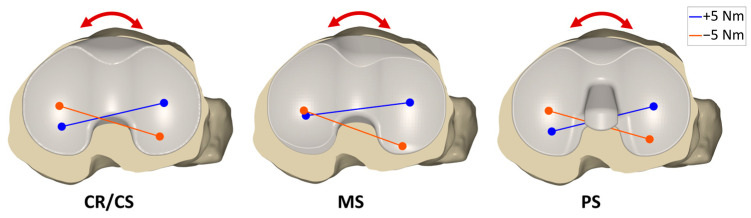
Projections of the flexion axis and the centers of the medial and lateral condyles of the femoral component onto the tibial plane with ±5 Nm internal (−) and external (+) rotation moment applied on the tibia at 45° flexion, showing the condylar motion with a CR/CS, MS, and PS TKA design exemplary for one specimen. The colors represent the respective moment on the tibia. Blue = +5 Nm; orange = −5 Nm.

**Figure 7 bioengineering-12-00087-f007:**
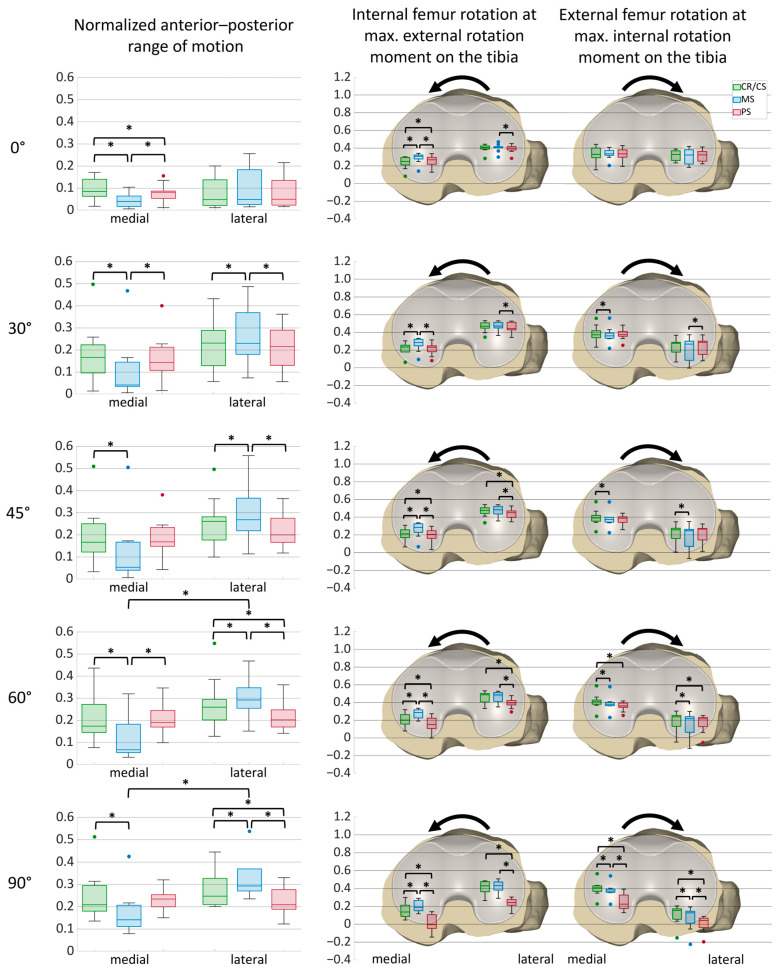
Anterior–posterior range of motion during internal–external rotation moments at different flexion angles and anterior–posterior positions of the medial and lateral femoral condyles at maximum internal and external rotation moments, respectively, at different flexion angles on a normalized tibia (*n* = 10). The anterior–posterior range of motion and the positions are normalized to an anterior–posterior tibia width of 1. Significant differences are marked with an asterisk (*p* ≤ 0.05). Internal femur rotation = external rotation moments on the tibia; external femur rotation = internal rotation moments on the tibia. Green = CR/CS; blue = MS; red = PS.

## Data Availability

The data presented in this study are available on request from the corresponding author. The data are not publicly available due to ethical and privacy considerations associated with human cadaveric donor material.

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
