# Peer review of "Constraint of Different Knee Implant Designs Under Anterior–Posterior Shear Forces and Internal–External Rotation Moments in Human Cadaveric Knees"

_bioengineering, 2025, doi:10.3390/bioengineering12010087_

Round 1

Reviewer 1 Report

Comments and Suggestions for Authors

First of all, I would like to thank the authors and congratulate them and future readers with a new brilliant work. I consider the obtained results as a significant contribution to our knowledge in the biomechanics of the TKA of different designs. It continues the recently published papers of the authors, which were obtained by their experimental technique and setup.
However, I have a question. In Figures 4 and 6 on the right side, I can see the dots positioned too far from the medial position, even out of the inlay. Maybe this is okay because the dots represent outliers. But I do not see such dots (outliers) in the range of motion on the left side of the figures. It is especially pronounced for the lateral condyle and for flexion angles of 45° and 90°. How  can this be? Because range of motion is measured in angles? I guess this should be discussed in the text.

Reviewer 2 Report

Comments and Suggestions for Authors

The author investigated the constraint of three different implant designs (symmetrical implants with and without a post-cam mechanism, and an asymmetrical medial-stabilized implant) under anterior-posterior shear forces and internal-external rotation moments at different flexion angles in human cadaveric knees using a six-degrees-of-freedom joint motion simulator. I think the paper can be considered with major revision after the following points are revised.

1. How does the medial-stabilized implant's unique design contribute to its reduced anterior-posterior translation medially compared to the symmetrical designs?

2. What are the potential clinical implications of the post-cam implant design's significantly more posterior position of the femoral condyles in flexion?

3. How do the differences in ligament situations and individual implant positioning affect the overall performance and stability of each implant design?

4. In what ways do the findings of this study advance our understanding of implant constraint under clinically relevant conditions?

5. How might the identified characteristics of each implant design influence future developments in total knee arthroplasty implants?

6. English should be improved.

Comments on the Quality of English Language

English should be improved.

Reviewer 3 Report

Comments and Suggestions for Authors

Comments: In this paper, the authors conducted a detailed analysis of three different implant designs for total knee arthroplasty, focusing on shear forces and rotation moments across various flexion angles in human cadaveric knees. The study aims to assist surgeons in selecting the most appropriate implant design tailored to specific patient needs. The findings from this paper are meaningful and will undoubtedly benefit medical professionals and patients alike. However, after carefully examining the paper, several issues were identified and need to be addressed:

1. In section 1 "Introduction", please provide a proper citation for the statement "In contrast, biomechanical in vitro studies offer high accuracy and control, as well as the ability to measure the constraint of different implant de- signs within the same ligamentous situation" (lines 57-59).

2. In subsection 2.1 "Specimen Preparation", more details are needed to clarify the procedures described. For example, for the preparation work described in the second paragraph, what does the obtained 3D fitting information look like? Also, which software was used for segmentation from the CT scan? Please provide a flowchart illustrating the procedures step by step, including the tools used in each step. For the figure illustration, clearly indicate the 0 degree tibial slope and the 3 degree posterior tibial slope, as described in lines 107-108: "The components were implanted using mechanical alignment and 0° tibial slope, whereas a posterior tibial slope of 3° is integrated in the inlay designs."

3. Please include a flowchart to clarify the procedures described in subsection 2.2. "Experimental Testing". The current description is difficult to understand without proper graphical representation.

4. For the Wilcoxon-signed rank test mentioned in lines 164-166: "Wilcoxon-signed rank tests were used to compare the position at maxi-mum anterior/posterior shear force and internal/external rotation moment, respectively, between the different TKA designs pairwise at various flexion angles", please provide proper citation.

5. For the results presented in Section 3, please explain the meaning and units of the values on the y-axis for Figure 4 (0~0.5/-0.4~1.2) and Figure 6 (0~0.6/-0.4~1.2). For the boxplots in Figures 4 and 6, consider merging them into an integrated boxplot to provide a comprehensive view that better supports the conclusions of the paper.

Round 2

Reviewer 2 Report

Comments and Suggestions for Authors

The paper quality is improved after the revision. Some key problems are explained reasonably and revised the manuscript accordingly. Now, I recommend the manuscript to be published in its current form.

Comments on the Quality of English Language

N/A